# Clinical Features of Acute Ischemic Stroke Patients with Hypoesthesia as an Initial Symptom

**Takayoshi Akimoto** , **Katsuhiko Ogawa, Makoto Hara, Satoko Ninomiya, Masaki Ishihara, Akihiko Morita, Satoshi Kamei and Hideto Nakajima ***

Division of Neurology, Department of Medicine, Nihon University School of Medicine, Tokyo 173-8610, Japan
* Correspondence: nakajima.hideto@nihon-u.ac.jp; Tel.: +81-3-3972-8111

**Abstract:** This study aimed to evaluate the clinical characteristics of acute ischemic stroke (AIS) patients who experienced hypoesthesia as the initial symptom. We retrospectively analyzed the medical records of 176 hospitalized AIS patients who met our inclusion and exclusion criteria and evaluated their clinical features and MRI findings. Among this cohort, 20 (11%) patients presented with hypoesthesia as the initial symptom. MRI scans of these 20 patients identified lesions in the thalamus or pontine tegmentum in 14 and brain lesions at other sites in 6. The 20 hypoesthesia patients had higher systolic ($p = 0.031$) and diastolic blood pressure ($p = 0.037$) on admission, and a higher rate of small-vessel occlusion ($p < 0.001$) than patients without hypoesthesia. The patients with hypoesthesia had a significantly shorter average hospital stay ($p = 0.007$) but did not differ significantly from those without hypoesthesia in National Institutes of Health Stroke Scale scores on admission ($p = 0.182$) or the modified Rankin Scale scores for neurologic disability on discharge ($p = 0.319$). In the patients with acute onset hypoesthesia, high blood pressure, and neurological deficits were more likely to be due to AIS than other causes. Since most of the lesions in AIS patients with hypoesthesia as the initial symptom were found to be small, we recommend performing MRI scans with such patients to confirm AIS.

**Keywords:** acute ischemic stroke; high blood pressure; hypoesthesia; MRI; sensory stroke; small-vessel occlusion

## 1. Introduction

Sensory disturbances are not uncommon in AIS. A 2015 study by the Japanese Stroke Databank found that the sensory disturbance rate in patients with AIS is 7.42% [1]. Sensory disturbances are often experienced on one side of the face and/or body and can range from mild to severe depending on the location and extent of brain damage. Sensory disturbances due to stroke include cheiro-oral syndrome (COS), which affects the perioral area and unilateral fingers and/or hand [2–7]; cheiro-pedal syndrome (CPS), which affects unilateral fingers and/or hand and ipsilateral foot and/or toes [8]; cheiro-oral-pedal syndrome (COPS), which affects the perioral area, unilateral fingers, and ipsilateral foot and/or toes [9,10]; restricted acral sensory syndrome (RASS), which affects peripheral areas [11]; and pure sensory stroke (PSS), which causes persistent or transient dysesthesia or paresthesia and mild unilateral whole body hypoesthesia without other neurological deficits [12]. Lesions in the cerebral cortex, corona radiata, internal capsule, thalamus, midbrain, pons, and medulla have been identified with these sensory disturbances [2–13]. In some cases, the sensory disturbances can decrease fine motor skills, causing difficulty with daily activities such as buttoning a shirt or handwriting. They can also affect balance and increase the risk of falls. Treatment for sensory disturbances due to AIS typically involves rehabilitation and physical therapy to help improve function and manage any residual numbness or weakness. In severe cases, medication or other medical interventions may be necessary. It is important for healthcare teams working with AIS survivors to

develop comprehensive, individualized treatment plans for each patient to help manage sensory disturbances and improve quality of life.

Sensory disturbances that result from AIS or other neurological conditions may be classed as subjective sensory disorders (SSD) or objective sensory disorders (OSD). SSD are those in which the patients' reports of changes in sensation are not accompanied by any observable physical signs. In these cases, patients may feel tingling or burning in a limb but with no visible or measurable abnormality in the limb. OSD are disturbances in which physical signs of the disturbances can be observed and measured. OSD patients may have reduced sensitivity to touch in a limb due to abnormalities in superficial or deep sensation. Sensory disturbances can be early indicators of a stroke, and prompt evaluation and treatment can improve outcomes. Early diagnosis and proper management of sensory disturbances during initial AIS treatment can help improve prognosis and reduce the risk of long-term complications, such as weakness, paralysis, or difficulty with activities of daily living.

In the previous AIS studies, some research studies distinguished between SSD and OSD [2,12] but some did not [3,11,13]. This is presumably because it is difficult to distinguish between SSD and OSD in clinical practice. The presentation of SSD varies among patients and includes dysesthesias and paresthesias. In this paper, we use the umbrella term "hypoesthesia" to describe all SSD symptoms. The severity of AIS in patients with hypoesthesia may be underestimated because these patients tend to show fewer apparent neurological abnormalities. Accurate evaluation and prompt treatment of these AIS patients with hypoesthesia should be a priority in AIS practice. Therefore, the current study aimed to analyze the clinical characteristics of AIS patients with hypoesthesia as the initial symptom.

## 2. Materials and Methods

### 2.1. Inclusion Criteria

There were 302 consecutive patients with AIS admitted to Nihon University Itabashi Hospital between 26 October 2015 and 26 October 2017. Patients who could not be adequately evaluated due to dementia, consciousness disturbance, severe aphasia, or unilateral spatial neglect were excluded. We enrolled the remaining patients retrospectively. To identify those with hypoesthesia, we evaluated the chief complaint and current medical history in the medical records of each. This study was approved by the Institutional Research Review Board of Nihon University (RK-150714-6).

### 2.2. Collected Data

The following information was collected from the medical records of patients who met our inclusion criteria: the presence or absence of hypoesthesia, age, gender, body mass index (BMI), systolic blood pressure (SBP), and diastolic blood pressure (DBP) on admission, medical history (hypertension, dyslipidemia, diabetes mellitus, stroke (ischemic/hemorrhagic), and hemodialysis for chronic kidney disease), blood sugar, HbA1c levels, C-reactive protein (CRP) levels, creatinine levels, low-density lipoprotein cholesterol (LDL-Cho) levels, National Institutes of Health Stroke Scale (NIHSS) scores, length of hospital stay, and discharge destination (home or other). A patient's medical history was taken into account if something pertinent was noted before admission or the patient was on medication at the time of admission (e.g., patients with a current prescription for antihypertensive medication were considered to have pre-existing hypertension). In addition, the presence or absence of pain due to the stroke was noted.

### 2.3. Neuroimaging and the Cause of Ischemic Stroke

All patients underwent brain computed tomography (CT) or brain magnetic resonance imaging (MRI) and magnetic resonance angiography (MRA) on the day of admission or the day after. MRI examinations were repeated in patients whose first MRI detected no responsible lesions. To determine the type of stroke, all patients underwent carotid echocar-

diography, transthoracic echocardiography, and 24-h continuous electrocardiography. The causes of strokes were categorized into large-artery atherosclerosis (LAA), cardio-embolism (CE), small-vessel occlusion (SVO), strokes of other determined etiology, and strokes of undetermined etiology (UD), according to the Trial of Org 10172 in the Acute Stroke Treatment classifications [14]. Specifically, if a cardiac source of emboli was confirmed, the cause was classified as CE. If there was more than 50% stenosis by MRA or carotid echocardiography in the region perfusing the lesion, it was classified as LAA. If LAA criteria were not met and the infarct was less than 1.5 cm in size, it was classified as SVO. If none of these criteria applied or more than one cause was applicable, the cause was classified as UD.

### 2.4. Statistical Analysis

Statistical analyses were performed using SPSS for Windows v. 18 (SPSS Inc., Chicago, IL, USA). Continuous variables were compared using the Mann–Whitney U test. Categorical variables were compared using Fisher's exact test. A *p*-value < 0.05 was considered statistically significant.

## 3. Results

### 3.1. Patient Characteristics

During the period of the study, 302 AIS patients were admitted to our department. Of these, 27 were excluded due to unclear or incomplete head CT or head MR images. Then, 99 were excluded due to no or unknown hypoesthesia evaluation. A total of 176 patients with acute ischemic lesions were included in the study and evaluated for hypoesthesia. We identified 20 (11.4%) patients with hypoesthesia as the initial symptom in whom the responsible lesions were detected (Figure 1). In this population, 17 were male and 3 were female and the median age was 66.5 years (49–82).

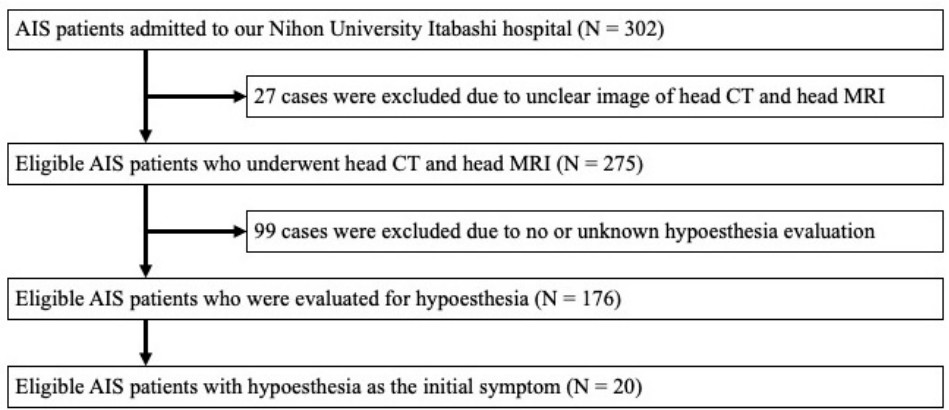

**Figure 1.** Study flow chart.

The profiles of the 20 AIS patients with hypoesthesia are shown in Table 1. As the chief complaint, seven (35%) patients reported only hypoesthesia (patients 2, 3, 6, 7, 9, 12, and 18), while the other 13 (65%) complained of additional symptoms. Including the seven patients whose chief complaint was hypoesthesia alone (patients 2, 3, 6, 7, 9, 12, and 18), other neurological deficits were detected in all 20 patients. These were facial palsy in four, dysarthria in eight, mild weakness of the limbs in ten (seven with monoplegia and three with hemiplegia of the arm), and hemiataxia in ten. OSD were confirmed by neurological examination in 10 patients. Impairment of superficial (tactile and/or pain) sensation was found in 10 patients, and impairment of vibratory sensation was found in 2 of the 12 patients whose vibratory sensation was examined. The severity of sensory decreases in OSD was generally mild, and none of the patients showed anesthesia. Motor weakness and ataxia were observed on the side ipsilateral to the ischemic lesion in one patient with a lateral medulla infarction (patient 20).

**Table 1.** Profiles of AIS patients with hypoesthesia as the initial symptom.

| | Age/Sex | Hypoesthesia | Neurological Symptoms | | | | | | MRI Lesion | Disease Type |
|---|---|---|---|---|---|---|---|---|---|---|
| | | | FP | DA | Weakness | Ataxia | DSS | DVS | | |
| 1 | 65/M | face | + | + | arm | arm/leg | − | − | thalamus | UD |
| 2 | 54/F | hemibody | − | − | − | arm/leg | − | − | thalamus | SVO |
| 3 | 67/M | lip/cheek/finger | − | − | arm | − | face/arm | NA | thalamus | SVO |
| 4 | 82/M | wrist-finger | − | + | arm | arm/leg | − | − | thalamus | SVO |
| 5 | 79/M | hemibody | − | − | arm/leg | − | face/arm/leg | − | thalamus | SVO |
| 6 | 71/M | cheek | − | − | − | − | face | NA | thalamus | SVO |
| 7 | 69/M | face/arm/leg | − | − | − | − | face/arm/leg | - | thalamus | SVO |
| 8 | 61/M | face/arm/leg | − | + | arm/leg | arm/leg | face/arm/leg | leg | thalamus | LAA |
| 9 | 70/M | hand/leg | − | − | − | arm | arm | − | PT | SVO |
| 10 | 72/M | palm/lower leg | − | + | − | arm/leg | − | − | PT | SVO |
| 11 | 66/M | hand/foot | − | − | − | arm | − | NA | PT | SVO |
| 12 | 49/M | forearm-finger/knee-toe | + | − | − | − | arm/leg | leg | PT | SVO |
| 13 | 65/M | hemibody | − | − | arm | - | face/arm/leg | NA | PT | SVO |
| 14 | 75/M | wrist-finger/sole | − | + | arm/leg | arm/leg | − | − | PB/PT | UD |
| 15 | 62/F | intraoral/throat/cheek | − | + | − | − | face | NA | putamen | UD |
| 16 | 62/F | forearm | − | − | arm | − | − | NA | corona radiata | SVO |
| 17 | 79/M | finger | + | + | − | − | − | NA | PLIC | SVO |
| 18 | 66/M | forearm-finger | − | − | arm | − | arm | NA | cerebral cortex | CE |
| 19 | 71/M | wrist-finger/knee-toe | + | + | − | arm/leg | − | − | MM | LAA |
| 20 | 64/M | hand * | − | − | arm * | arm/leg * | − | − | LM | SVO |

FP: facial palsy, DA: dysarthria, DSS: diminished superficial sensation, DVS: diminished vibratory sensation, NA: not available, PT: pontine tegmentum, PB: pontine base, PLIC: posterior limb of the internal capsule, MM: medial medulla, LM: lateral medulla, UD: undetermined etiology, SVO: small-vessel occlusion, LAA: large-artery atherosclerosis, CE: cardio-embolic. * The subjective sensory disturbances, limb weakness, and ataxia of patient 20 involved the side ipsilateral to the ischemic lesion.

*3.2. Lesions and the Distribution of Hypoesthesia*

Diffusion-weighted brain images of the 20 patients with hypoesthesia are shown in Figure 2. The thalamus was the most commonly affected region ($n = 8$; patients 1–8), followed by the pontine tegmentum ($n = 5$; patients 9–13), pontine base to tegmentum ($n = 1$; patient 14), putamen ($n = 1$; patient 15), corona radiata ($n = 1$; patient 16), posterior limb of the internal capsule ($n = 1$; patient 17), cerebral cortex ($n = 1$; patient 18), medial medulla ($n = 1$; patient 19), and lateral medulla ($n = 1$; patient 20). Of the 19 patients who underwent brain CT, 15 showed no acute lesions on that scan.

The hypoesthesia was contralateral to the lesion in 19 patients (except for patient 20). As shown in Table 1, the hypoesthesia was distributed across the face, cheek, or throat in three patients and was caused by thalamic (patients 1 and 6) or putaminal (patient 15) infarction. Hypoesthesia of the arm was caused by lesions in the thalamus (patient 4), corona radiata (patient 16), posterior limb of the internal capsule (patient 17), parietal cerebral cortex (patient 18), or lateral medulla (patient 20). COS was seen in one patient with thalamic infarction (patient 3). CPS was caused by lesions in the pontine tegmentum in four patients (patients 9, 10, 11, and 12), the pontine base and tegmentum in one patient (patient 14), and the medial medulla in one patient (patient 19). COPS was seen in two patients, both with thalamic infarctions (patients 7 and 8). PSS was seen in three patients, two with thalamic infarctions (patients 2 and 5) and one with a pontine tegmentum infarction (patient 13). In addition, none of the 176 patients included in the study developed central neuropathic pain during the observation period.

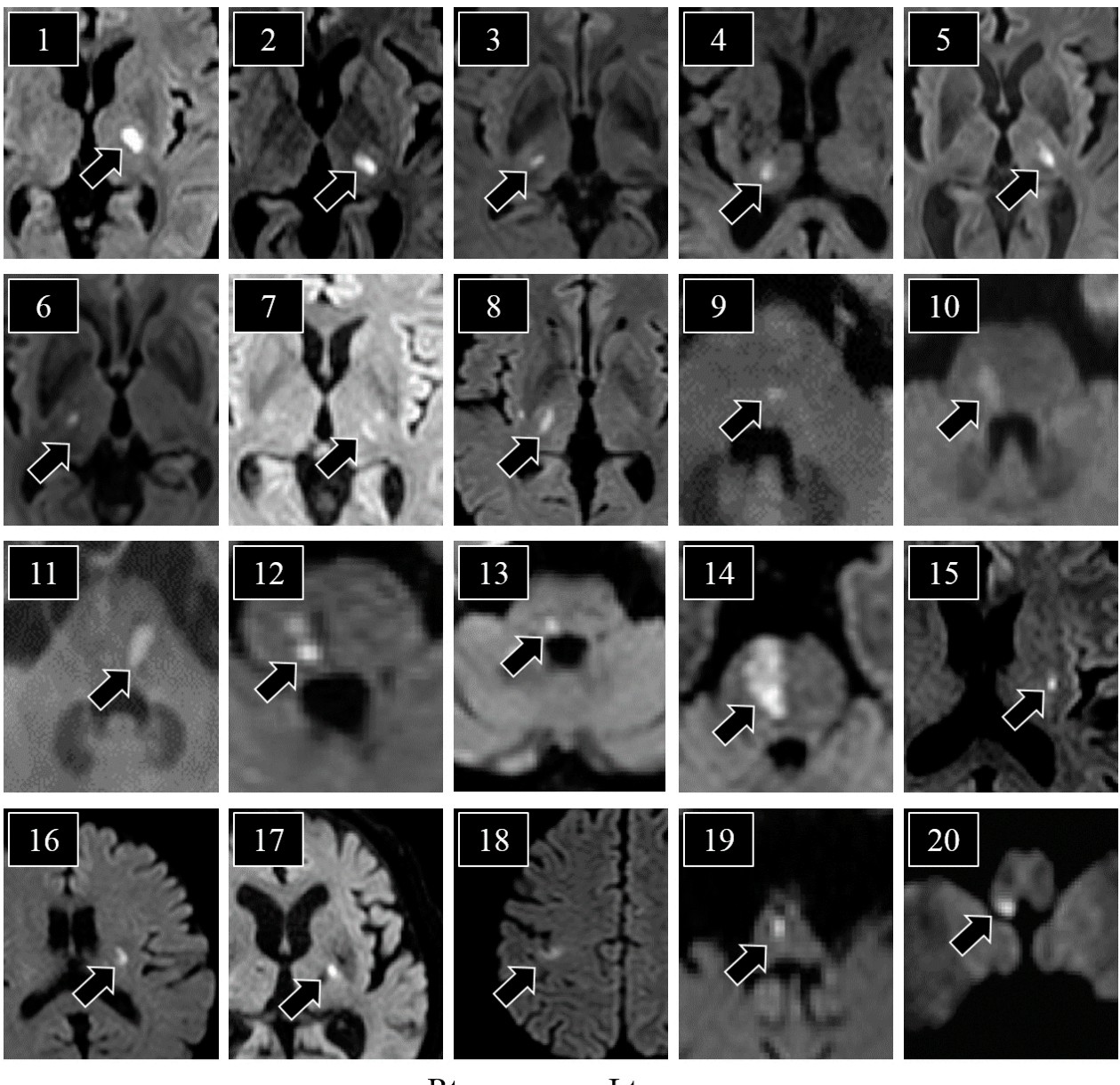

Rt ⟷ Lt

**Figure 2.** Diffusion-weighted magnetic resonance imaging of patients with subjective sensory disturbances/hypoesthesia. The black arrowheads show the ischemic lesions. Lesions were detected in the thalamus (*n* = 8; patients 1–8), pontine tegmentum (*n* = 5; patients 9–13), pontine base to tegmentum (*n* = 1; patient 14), putamen (*n* = 1; patient 15), corona radiata (*n* = 1; patient 16), posterior limb of the internal capsule (*n* = 1; patient 17), cerebral cortex (*n* = 1; patient 18), medial medulla (*n* = 1; patient 19), and lateral medulla (*n* = 1; patient 20). Lt: left, Rt: right.

*3.3. Comparison of Patients with and without Hypoesthesia*

We divided the AIS patients into two groups—those with and without hypoesthesia—and then compared the clinical profiles and outcomes of these two groups (Table 2). There were no significant between-group differences in age or sex (*p* = 0.160, *p* = 0.177). However, systolic (*p* = 0.031) and diastolic (*p* = 0.037) blood pressure on admission were both significantly higher in patients with hypoesthesia.

**Table 2.** Characteristics of the AIS patients with and without hypoesthesia.

| Characteristics | Total (*n* = 176) | With Hypoesthesia (*n* = 20) | Without Hypoesthesia (*n* = 156) | *p*-Value |
|---|---|---|---|---|
| Physical examination | | | | |
| Age (y.o.) | 71 (39–91) | 66.5 (49–82) | 71 (39–91) | 0.160 |
| Sex (Male), *n* (%) | 130 (73.8%) | 17 (85%) | 113 (72.4%) | 0.177 |
| SBP on admission (mmHg) | 158 (92–264) | 171.5 (128–230) | 156 (92–264) | * 0.031 |
| DBP on admission (mmHg) | 89 (33–132) | 99 (63–125) | 89 (33–132) | * 0.037 |
| Body mass index (kg/m$^2$) | 23.1 (14.5–34.4) | 23 (14–33) | 24 (18–34) | 0.268 |
| NIHSS on admission | 2 (0–10) | 2 (1–5) | 2 (0–10) | 0.182 |
| Past medical history | | | | |
| Hypertension, *n* (%) | 106 (61%) | 8 (42%) | 98 (63%) | 0.064 |
| Dyslipidemia, *n* (%) | 53 (30%) | 6 (30%) | 47 (30%) | 0.607 |
| Diabetes mellitus, *n* (%) | 52 (30%) | 6 (30%) | 46 (29%) | 0.573 |
| Hemodialysis, *n* (%) | 7 (4%) | 0 (0%) | 7 (4%) | 0.423 |
| Stroke (ischemic/hemorrhagic), *n* (%) | 38 (22%) | 3 (15%) | 35 (22%) | 0.332 |
| Laboratory data | | | | |
| Blood sugar (mg/dL) | 119 (76–337) | 118 (76–332) | 127 (88–337) | 0.402 |
| HbA1c (%) | 6.1 (4.4–13.6) | 6.1 (4.4–13.6) | 6.2 (4.9–10.8) | 0.531 |
| LDL-Cho (mg/dL) | 122 (31–242) | 123 (49–242) | 110 (31–183) | 0.119 |
| Creatinine (mg/dL) ** | 0.78 (0.42–9.00) | 0.77 (0.42–9.00) | 0.87 (0.52–2.35) | 0.311 |
| CRP (mg/dL) | 0.15 (0.01–30.0) | 0.15 (0.01–30.0) | 0.11 (0.06–0.56) | 0.771 |
| NT-pro BNP (pg/mL) ** | 174 (10–11,222) | 174 (10–11,222) | 151 (10–8462) | 0.211 |
| Disease type | | | | |
| Large-artery atherosclerosis, *n* (%) | 20 (11.3%) | 2 (10%) | 18 (11.5%) | 0.596 |
| Cardio-embolism, *n* (%) | 43 (24.4%) | 1 (5%) | 42 (26.9%) | * 0.021 |
| Small-vessel occlusion, *n* (%) | 47 (26.7%) | 14 (70%) | 33 (21.2%) | * <0.001 |
| Undetermined etiology, *n* (%) | 66 (37.5%) | 3 (15%) | 63 (40.4%) | * 0.021 |
| Prognosis | | | | |
| mRS on discharge | 1 (0–6) | 1 (0–3) | 1 (0–6) | 0.319 |
| Duration of hospitalization (days) *** | 19 (8–104) | 15.5 (8–24) | 20 (9–104) | * 0.007 |
| Discharge to home | 127 (73%) | 15 (75%) | 112 (73%) | 0.534 |

Data for the continuous variables are presented as median (range). The *p*-values for continuous variables were calculated using Mann-Whitney U tests. The *p*-values for categorical variables were calculated using Fisher's exact tests. * *p* < 0.05, ** excluding hemodialysis cases, *** excluding cases of death. BMI, body mass index; CRP, C-reactive protein; DBP, diastolic blood pressure; LDL-Cho, low-density lipoprotein cholesterol; mRS, modified Rankin scale; NIHSS, National Institutes of Health Stroke Scale; SBP, systolic blood pressure.

The most common cause of AIS in those with hypoesthesia was small-vessel occlusion (*n* = 14), followed by UD, (*n* = 3), LAA (*n* = 2), and CE (*n* = 1). SVO was significantly more common (*p* < 0.001), and cardio-embolism was significantly less common (*p* = 0.021) in patients with hypoesthesia than in those without but there was no significant difference in the rate of LAA between groups (*p* = 0.596).

The NIHSS on admission ranged from 1 to 5 points (median, 2 points) in the AIS patients with hypoesthesia. There was no significant difference in the NIHSS score between patients with and without hypoesthesia (*p* = 0.182). In the hypoesthesia group, the modified Rankin scale (mRS) scores on discharge were 0 (*n* = 6), 1 (*n* = 8), 2 (*n* = 5), and 3 (*n* = 1). There was no significant between-group difference in discharge mRS scores (*p* = 0.319). However, the duration of hospitalization in the hypoesthesia patients ranged from 8 to 24 days (median 15.5 days), which was significantly shorter than that of the patients without hypoesthesia (9–104 days; median, 20 days) (*p* = 0.007).

## 4. Discussion

We retrospectively analyzed the 176 consecutive hospitalized AIS patients, 11% of whom presented with hypoesthesia as the initial symptom. The hypoesthesia patients had higher systolic and diastolic blood pressure on admission, and more lesions of the

thalamus and pontine tegmentum caused by small-vessel occlusion than patients without hypoesthesia. The results suggest that high blood pressure and neurological deficits in patients with acute onset hypoesthesia indicate AIS as a cause. We also found that the AIS patients with hypoesthesia had a shorter hospital stay, although the NIHSS on admission in the AIS patients with hypoesthesia did not differ from those without hypoesthesia.

In reviews and case series on sensory stroke, the mechanism of hypoesthesia due to stroke has not been adequately discussed [2,3,5,6,8,11,13]. Ochoa et al. have shown hypoesthesia in peripheral neuropathy to be caused by ectopic impulses in the nerve fibers. In their study, the upper arms of healthy participants were put in an ischemic state by manchette compression, and the relationships between the symptoms that appeared after ischemic release and the ulnar nerve microelectrodes were recorded. Symptoms such as buzzing, tingling, pricking, and pseudo-cramp were observed in association with the ectopic impulse, which was recorded by electrogram [15]. When peripheral nerves suffer ischemia, both the adenosine triphosphate (ATP) content in the nerve axon and the activity of the Na-K pump decrease, resulting in membrane depolarization [16,17]. The action potential threshold is altered by the changes in pump activity [16,17]. In sensory nerves, this threshold is lowered and then raised [18]. This change in threshold produces the impulses responsible for hypoesthesia [16–18]. Patients who receive artificial spinal cord stimulation are aware of paresthesia in the area corresponding to the stimulated area [19]. It is reasonable to assume that the hypoesthesia caused by brain lesions is also due to ectopic impulses.

In the present study, no significant differences were found between the age, sex, BMI, NIHSS scores, medical history, or laboratory data of AIS patients with and without hypoesthesia. However, AIS patients with hypoesthesia had significantly higher blood pressure on admission. Although blood pressure is generally elevated in AIS patients, such abnormal hypertension associated with hypoesthesia is a reason to suspect AIS. In particular, SVOs are associated with higher blood pressure than non-SVOs [20]. Previous research on sensory stroke had also found SVO to be the most common cause of AIS [2,13]. Among our AIS patients with hypoesthesia, the most common disease type was also SVO. The lesions in both SVO and non-SVO cases were small, so the lesion detection rate on CT examination on admission was low (5/19 patients; 26.3%). Therefore, patients with hypoesthesia suspected of having AIS should undergo an MRI.

Of the 20 AIS patients with hypoesthesia, 7 experienced no symptoms other than the hypoesthesia. However, other neurological deficits were found in all 20, including dysarthria, ataxia, and motor weakness. As these are all common in AIS, they were not useful for determining the locations of the ischemic lesions. Some of the patients had impaired deep sensation, some had impaired superficial sensation, and some had neither. It can be assumed that a part of the deep sensory pathway or the pain sensory pathway was impaired in the patients with hypoesthesia, indicating that the pathways impaired in hypoesthesia due to AIS may vary. In the AIS patients with hypoesthesia, the lesion locations were the cerebral cortex, the corona radiata, the internal capsule, the thalamus, the midbrain, the pons, and the medulla. These are almost identical to the lesion locations previously found to cause COS, CPS, COPS, RASS, and PSS [2–11,13,21]. We found no correlations between the locations of the ischemic lesions and the distribution of hypoesthesia or concomitant neurological deficits, suggesting that it would be difficult to predict the correct locations of lesions from the distribution of hypoesthesia.

There was no significant difference in mRS scores at discharge between patients with and without hypoesthesia. Typically, patients with gait disturbance require inpatient rehabilitation until they can ambulate independently, and a longer hospitalization period enables them to achieve an mRS score similar to that of patients without gait disturbance. Patients with hypoesthesia had significantly shorter hospital stays than those without hypoesthesia. This suggested that the residual aftereffects of hypoesthesia had a lesser impact on gait and did not necessitate prolonged hospital rehabilitation.

In addition to hypoesthesia, sensory impairments after stroke can also include pain. Neuropathic pain caused by cerebral lesions is known as central poststroke pain (CPSP) [22–27]. CPSP occurs in up to 35% of patients after cerebral hemorrhage or ischemic stroke and results from lesions in the thalamus, brainstem, basal ganglia, and insula [22,24–27]. The thalamus is the most common lesion site in patients who develop CPSP [22] and CPSP-associated thalamic lesions occur more often in the lateral and posterior than the medial thalamus [23]. In the present study, the lesions were found in areas where CPSP could occur, but we saw no cases of CPSP during the observation period (8–104 days of hospitalization) either in those with hypoesthesia or those without. While a previous report found diabetes to increase the risk of CSPS in stroke patients [26], we found no significant association between AIS patients with hypoesthesia and a medical history of diabetes mellitus, blood sugar levels, or HbA1c titers. CSPS can appear several days to several months after stroke onset [26,28,29]. The association between hypoesthesia and CPSP is unclear, but this may become more evident with long-term follow-up. Therefore, the long-term prognosis of AIS cases with hypoesthesia is a question for future research.

The present study had some limitations. The present study design was retrospective and our AIS with hypoesthesia sample size was too small to perform multivariate analysis. Patients admitted to our department with hypoesthesia whose lesions were detected by MRI examination were evaluated; however, not all patients with acute-onset hypoesthesia underwent brain MRI examinations. Some patients with AIS might be misdiagnosed with spondylosis or neuropathy. Rapid MRI examination would yield further clinical information about patients with acute-onset hypoesthesia. The evaluation of hypoesthesia as SSD in AIS patients may not be important in clinical practice because of the difficulty of specifically assessing hypoesthesia. However, the results of the present study indicate that assessment of hypoesthesia could contribute to the accurate evaluation and prompt treatment of the AIS patients.

## 5. Conclusions

The combination of acute onset unilateral hypoesthesia with high blood pressure and mild neurological abnormalities can suggest the presence of AIS. While the correct locations of lesions could not be accurately determined based on the distribution of hypoesthesia in patients with AIS, their lesions were mainly located in the thalamus and pontine tegmentum and were usually caused by small-vessel occlusion. AIS patients presenting with hypoesthesia had shorter hospital stays. Since most lesions in AIS patients with hypoesthesia as the initial symptom are small, an MRI should be performed to confirm AIS.

**Author Contributions:** All authors contributed to the conception and design of this study. Material preparation, data collection, and statistical analysis were performed by T.A., K.O., S.N., M.H., M.I. and A.M.; T.A. wrote the first manuscript; K.O., S.K. and H.N. helped draft the manuscript; and all authors critically revised the manuscript. All authors have read and agreed to the published version of the manuscript.

**Funding:** This research was partly supported by JSPS KAKENHI (grant No. JP 21K17692).

**Institutional Review Board Statement:** This study was conducted according to the tenets of the 2013 revision of the Declaration of Helsinki and was approved by the Institutional Research Review Board of Nihon University (RK-150714-6).

**Informed Consent Statement:** Informed consent was obtained from the patients whose data were evaluated. No individual participants are identified in the publication.

**Data Availability Statement:** The data that support the findings of this study are available on reasonable request from the corresponding author.

**Acknowledgments:** The authors would like to thank Hiroko Minami at Amekudai Hospital, Hiroshi Shiota and Mari Saito at Kawaguchi Municipal Medical Center, Keiji Shiobara at Nagaoka-Nishi Hospital, and Kazutaka Mitsuke at Nihon University Hospital for their help collecting patient data.

**Conflicts of Interest:** The authors declare no conflict of interest.

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
