# Peer review of "Clinical Features of Acute Ischemic Stroke Patients with Hypoesthesia as an Initial Symptom"

_2035-8377, doi:10.3390/neurolint15010030_

Round 1

Reviewer 1 Report (Previous Reviewer 3)

Dear Authors,

I am glad to have the opportunity to review your work. This study was aimed to evaluate the clinical characteristics of acute ischemic stroke (AIS) patients who experienced hypoesthesia as the initial symptom.

Adding new variables has significantly improved the quality of the paper and statistical analysis.

You have investigated only clinical features and in that sense descriptive statistics is correct. Thank you for removing prognosis from Conclusion since you have not investigated prognostic factors and you cannot conclude that based on your results. That was major flaw from previous version of the paper. Other corrections have significantly improved the quality of the paper, especially adding new references and improvement of the discussion.

Therefore, I recommend to accept the paper.

Author Response

Thank you very much for your careful peer review. We appreciate your assessment.

Reviewer 2 Report (New Reviewer)

Dear Authors,

thank you for providing the results of your work. I would suggest several corrections/clarifications:

1. line 103 - please confirm if this was blood-pressure on admission?

2. line 104 - please explain what do you mean by "hemodyalisis" 

3. line 114 - if you perform angiography of any kind and especially MRA several days after stroke, some vessel occlusion might be missed. Please put the timing of MRA more precisely, and also comment on this - in discussion part.  I suppose that etiology of stroke in your patients was not determined only by lack of large vessel occlusion, but by other characteristics of small vessel occlusion stroke on MRI. So please discuss it in more details

4. lines 230 - 243 and 284 - 293: I would suggest to skip the pathophysiology of pain in general, but to focus more on your own results in comparison with other known facts of this topic in the litterature.

5. line 259: this conclusion is not totally correct.  If a neurological deficit is common, it does not mean that it can not determine the location of the lesion. I would recommend to change this conclusion or to take it out.

6. line 281: generally wrong conclusion. Hypoesthaesia is not the cause, but the consequence of this type of (mild) sroke. Please reconsider/explain this conclusion

7. line 305: please check the grammar of this sentence

8. general recommendation - I would suggest to discuss in comparison with literature the importance of hypoesthesia as an initial symptom and use of MRI in patients with unusual high RR a little bit more, as this is the most important clinical message from your paper.

Best regards, wishing you a lot of success in your further work.

Author Response

We wish to thank the reviewer for the informative and supportive suggestion. We have addressed all the comments and modified the manuscript accordingly. Our responses to all the comments are provided in a point-by-point manner below. The corresponding changes in the revised manuscript are shown as highlighted text in the revised manuscript.

  1. line 103 - please confirm if this was blood-pressure on admission?

>> Thank you for your comment. Blood pressure was measured at the time of admission.

  • Line 83-

systolic blood pressure (SBP) and diastolic blood pressure (DBP) on admission

  1. line 104 - please explain what do you mean by "hemodyalisis"

>>We are sorry for the confusing content. This means maintenance hemodialysis for chronic kidney disease.

  • Line 85-

hemodialysis for chronic kidney disease

  1. line 114 - if you perform angiography of any kind and especially MRA several days after stroke, some vessel occlusion might be missed. Please put the timing of MRA more precisely, and also comment on this - in discussion part. I suppose that etiology of stroke in your patients was not determined only by lack of large vessel occlusion, but by other characteristics of small vessel occlusion stroke on MRI. So please discuss it in more details

>> We apologize that our description here was not accurate. In our hospital, MRI and MRA are taken at the same time. In all cases, CT or MRI was performed on the day of admission or the day after, so the phrase "several days" was ambiguous. In addition, carotid echocardiography, transthoracic echocardiography, and electrocardiography over 24 hours were performed in all patients to search for the cause of stroke during hospitalization. We added sentences about how we determined the subtype of stroke.

  • Line 94-

All patients underwent brain computed tomography (CT) or brain magnetic resonance imaging (MRI) and magnetic resonance angiography (MRA) on the day of admission or the day after.

  • Line 97-

To determine the type of stroke, all patients underwent carotid echocardiography, transthoracic echocardiography, and 24-hour continuous electrocardiography.

  • Line 102-

Specifically, if a cardiac source of emboli was confirmed, the cause was classified as CE. If there was more than 50% stenosis by MRA or carotid echocardiography in the region perfusing the lesion, it was classified as LAA. If LAA criteria were not met and the infarct was less than 1.5 cm in size, it was classified as SVO. If none of these criteria applied or more than one cause was applicable, the cause was classified as UD.

  1. lines 230 - 243 and 284 - 293: I would suggest to skip the pathophysiology of pain in general, but to focus more on your own results in comparison with other known facts of this topic in the litterature.

>> Thank you for your comment. We have removed the reference to pain. We also kept the part about post-stroke pain as it pertains to this study and removed the rest. While there is a lot of literature discussing the mechanism of post-stroke pain, there is not enough discussion about the hypoesthesia caused by stroke. We thought this point was important, so we left the description of the mechanism that produces hypoesthesia.

  • Line 206-

In reviews and case series on sensory stroke, the mechanism of hypesthesia due to stroke has not been adequately discussed [2,3,5,6,8,11,13].

  • Line 260-

In the present study, the lesions were found in areas where CPSP could occur, but we saw no cases of CPSP during the observation period (8–104 days of hospitalization) either in those with hypoesthesia or those without.

  1. line 259: this conclusion is not totally correct. If a neurological deficit is common, it does not mean that it can not determine the location of the lesion. I would recommend to change this conclusion or to take it out.

>> Thanks for pointing that out. We have removed this sentence.

  1. line 281: generally wrong conclusion. Hypoesthaesia is not the cause, but the consequence of this type of (mild) stroke. Please reconsider/explain this conclusion

>> Thank you very much for your comment. Although there was no significant difference in mRS at discharge between AIS with and without hypoesthesia, the former had a shorter hospital stay. This suggested that the stroke with hypoesthesia did not have sequelae that interfered with walking, and the patient could have been discharged home without requiring prolonged walking rehabilitation. The order of the sentences in this paragraph has been rearranged and the content rewritten as follows.

  • Line 229-

Among our AIS patients with hypoesthesia, the most common disease type was also SVO. The lesions in both SVO and non-SVO cases were small, so the lesion detection rate on CT examination on admission was low. (5/19 patients; 26.3%). Therefore, patients with hypoesthesia suspected of having AIS should undergo an MRI.

  • Line 247-

There was no significant difference in mRS scores at discharge between patients with and without hypoesthesia. Typically, patients with gait disturbance require inpatient rehabilitation until they can ambulate independently, and a longer hospitalization period enables them to achieve an mRS score similar to that of patients without gait disturbance. Patients with hypoesthesia had significantly shorter hospital stays than those without hypoesthesia. This suggested that the residual aftereffects of hypoesthesia had a lesser impact on gait, and did not necessitate prolonged hospital rehabilitation

  1. line 305: please check the grammar of this sentence

>> We apologize for the inappropriate sentence. This sentence has been revised as follows.

  • Line 269-

The present study design was retrospective and our AIS with hypoesthesia sample size was too small to perform multivariate analysis.

  1. general recommendation - I would suggest to discuss in comparison with literature the importance of hypoesthesia as an initial symptom and use of MRI in patients with unusual high RR a little bit more, as this is the most important clinical message from your paper.

>>Thank you for your constructive comments. It has been reported that marked high blood pressure is associated with the development of lacunar infarction, and in our study, hypoesthesia was often associated with marked hypertension and SVO. Therefore, we have stated that SVO should be suspected in cases of hypoesthesia with marked hypertension and that MRI is recommended in these cases due to the low detection rate of lesions on CT. The order of the sentences was changed along with the correction of Point 6.

  • Line 224-

However, patients with hypoesthesia had significantly higher blood pressure on admission. Although blood pressure is generally elevated in AIS patients, such abnormal hypertension associated with hypoesthesia is a reason to suspect AIS. In particular, SVOs are associated with higher blood pressure than non-SVOs [20]. Previous research on sensory stroke had also found SVO to be the most common cause of AIS [2,13]. Among our AIS patients with hypoesthesia, the most common disease type was also SVO. The lesions in both SVO and non-SVO cases were small, so the lesion detection rate on CT examination on admission was low. (5/19 patients; 26.3%). Therefore, patients with hypoesthesia suspected of having AIS should undergo an MRI.

Other corrections

Abbreviations were added for large-artery atherosclerosis (LAA), cardio-embolism (CE), and small-vessel occlusion (SVO).

The following text was added at the beginning of the second paragraph (Line 222-) “In the present study,”

This manuscript is a resubmission of an earlier submission. The following is a list of the peer review reports and author responses from that submission.

Round 1

Reviewer 1 Report

The statistical analysis only mentions the tests that were performed but does not show how the statistical significance of the study was verified under what criteria it is considered, in the summary it shows 178 patients and in the statistical analysis 176 meet the criteria, this in Figure 1 Study flow chart, and in paragraph 77.

Based on the findings, the prognosis of AIS patients with hypoesthesia was generally favorable. The writing of the conclusion should be improved considering only what was found in the investigation based on the results.

Author Response

The statistical analysis only mentions the tests that were performed but does not show how the statistical significance of the study was verified under what criteria it is considered, in the summary it shows 178 patients and in the statistical analysis 176 meet the criteria, this in Figure 1 Study flow chart, and in paragraph 77.

>> Thanks to the reviewer for the above comments. For the front part of the sentence, the statistical significance is set at P<0.05. This is shown in Line 81 of the revised manuscript. For the latter part of the sentence, we apologize for the error in the number of cases in the abstract section. The number of cases analyzed in this study was 176.

  • Line 10-

We retrospectively analyzed the medical records of 176 hospitalized AIS patients who met our inclusion and exclusion criteria and evaluated their clinical features and MRI findings.

Based on the findings, the prognosis of AIS patients with hypoesthesia was generally favorable. The writing of the conclusion should be improved considering only what was found in the investigation based on the results.

>>Thanks for pointing this out. As you pointed out, no statistically significant differences in outcome were found. Therefore, the sentences are changed as follows.

  • Line 20- We deleted the below sentence.

The prognosis of AIS patients with hypoesthesia was generally favorable.

  • Line 305-

AIS patients presenting with hypoesthesia had shorter hospital stays..

Reviewer 2 Report

The literature review, methodological aspects and conclusions seems appropriate.

In the description of clinical symptoms, however,  I noticed a lack of description of the painful conditions. For instance, we know that central neuropathic pain, a finding that may occur in up to 25% of patients with post-stroke somatossensory deficits, are classically described in thalamic ventro-lateral nuclei, thalamic-capsular or elsewhere in the somatossensory pathways, and may have an important clinical impact a subset of patients.

The inclusion of painful symptoms, if possible, would bring new elements of interest to the study. Anyway, as this was proposed to specifically assess a negative symptom (hypoesthesia), this gap do not compromise the quality and originality of the study.

Author Response

In the description of clinical symptoms, however, I noticed a lack of description of the painful conditions. For instance, we know that central neuropathic pain, a finding that may occur in up to 25% of patients with post-stroke somatossensory deficits, are classically described in thalamic ventro-lateral nuclei, thalamic-capsular or elsewhere in the somatossensory pathways, and may have an important clinical impact a subset of patients. The inclusion of painful symptoms, if possible, would bring new elements of interest to the study. Anyway, as this was proposed to specifically assess a negative symptom (hypoesthesia), this gap do not compromise the quality and originality of the study.

>>I appreciate your constructive comments. None of the 176 patients included in the study developed central neuropathic pain during the observation period (until discharge). However, CPSP is an important issue and we have added patient data and discussion.

  • Line 111-

In addition, the presence or absence of pain due to the stroke was noted.

  • Line 183-

In addition, none of the 176 patients included in the study developed central neuropathic pain during the observation period.

  • Line 272-

In addition to hypoesthesia, sensory impairments after stroke can also include pain. Pain produced by either external or internal stimulation of nociceptors is called nociceptive pain. The severity of nociceptive pain depends on the intensity and duration of the stimulus. Pain caused by damage to the nerves involved in somatosensory perception is called neuropathic pain. Neuropathic pain can occur in the absence of stimulation or, when it is a response to stimulation, may be inappropriately strong or persistent. Symptoms such as allodynia and hyperalgesia are types of neuropathic pain [21,23]. Neuropathic pain caused by cerebral lesions is known as central poststroke pain (CPSP) [24–29]. CPSP occurs in up to 35% of patients after cerebral hemorrhage or ischemic stroke and results from lesions in the thalamus, brainstem, basal ganglia, and insula [24,26–29]. The thalamus is the most common lesion site in patients who develop CPSP [24] and CPSP-associated thalamic lesions occur more often in the lateral and posterior than the medial thalamus [25]. In this study, we saw no cases of CPSP during the observation period (8–104 days of hospitalization) either in those with hypoesthesia or those without. While a previous report found diabetes to increase the risk of CSPS in stroke patients [28], we found no significant association between AIS patients with hypoesthesia and a medical history of diabetes mellitus, blood sugar levels, or HbA1c titers. CSPS can appear several days to several months after stroke onset [28,30,31]. The association between hypoesthesia and CPSP is unclear, but this may become more evident with long-term follow-up. Therefore, the long-term prognosis of AIS cases with hypoesthesia is a question for future research.

Reviewer 3 Report

Dear Authors,

I am glad to have the opportunity to review your work. This study was aimed to evaluate the clinical characteristics of acute ischemic stroke (AIS) patients who experienced hypoesthesia as the initial symptom.

The paper has interesting topic, however the power of the study is small. The patients were divided in two groups, whereas group with hypoesthesia was only 20 (11%). This is a small sample size for one of the outcomes. So, study design is not good. In this case the paper could have been presented as case-series, but not as cohort study. Also, the statistical analyses is not good. Only descriptive statistics was used. There is no univariate or multivariate logistic regression – probably due to small sample size. However, in this case, we cannot discuss about prognostic factors. It would have been interested if the authors did binomial regression.

However, there are major flaws – study design and statistical analyses.

Therefore, I recommend rejection of the paper.

Author Response

The paper has interesting topic, however the power of the study is small. The patients were divided in two groups, whereas group with hypoesthesia was only 20 (11%). This is a small sample size for one of the outcomes. So, study design is not good. In this case the paper could have been presented as case-series, but not as cohort study. Also, the statistical analyses is not good. Only descriptive statistics was used. There is no univariate or multivariate logistic regression – probably due to small sample size. However, in this case, we cannot discuss about prognostic factors. It would have been interested if the authors did binomial regression.

>>Thank you for pointing this out. The sample size of this study was small, which limited statistical analysis. Together with the results of this study, we hope to collect more cases that can be subjected to multivariate analysis in the future.

  • Line 293-

The study design was retrospective and our AIS with hypoesthesia sample size was small, validity and reliability of our statistical analyses.